# Evidence for Toxic Advanced Glycation End-Products Generated in the Normal Rat Liver

**DOI:** 10.3390/nu11071612

**Published:** 2019-07-16

**Authors:** Takanobu Takata, Akiko Sakasai-Sakai, Jun-ichi Takino, Masayoshi Takeuchi

**Affiliations:** 1Department of Advanced Medicine, Medical Research Institute, Kanazawa Medical University, Uchinada, Ishikawa 920-0293, Japan; 2Department of Biochemistry, Faculty of Pharmaceutical Sciences, Hiroshima International University, Kure, Hiroshima 737-0112, Japan

**Keywords:** toxic advanced glycation-end products (TAGE), high-fructose corn syrup (HFCS), *Lactobacillus* beverage, normal liver, serum levels of TAGE, intracellular TAGE, lifestyle-related diseases (LSRD), non-alcoholic fatty liver disease (NAFLD)

## Abstract

Glucose/fructose in beverages/foods containing high-fructose corn syrup (HFCS) are metabolized to glyceraldehyde (GA) in the liver. We previously reported that GA-derived advanced glycation end-products (toxic AGEs, TAGE) are generated and may induce the onset/progression of non-alcoholic fatty liver disease (NAFLD). We revealed that the generation of TAGE in the liver and serum TAGE levels were higher in NAFLD patients than in healthy humans. Although we propose the intracellular generation of TAGE in the normal liver, there is currently no evidence to support this, and the levels of TAGE produced have not yet been measured. In the present study, male Wister/ST rats that drank normal water or 10% HFCS 55 (HFCS beverage) were maintained for 13 weeks, and serum TAGE levels and intracellular TAGE levels in the liver were analyzed. Rats in the HFCS group drank 127.4 mL of the HFCS beverage each day. Serum TAGE levels and intracellular TAGE levels in the liver both increased in the HFCS group. A positive correlation was observed between intracellular TAGE levels in the liver and serum TAGE levels. On the other hand, in male Wister/ST rats that drank *Lactobacillus* beverage for 12 weeks—a commercial drink that contains glucose, fructose, and sucrose— no increases were observed in intracellular TAGE or serum TAGE levels. Intracellular TAGE were generated in the normal rat liver, and their production was promoted by HFCS, which may increase the risk of NAFLD.

## 1. Introduction

Total fat intake has decreased in the last decade in developed countries; however, the amounts of added glucose and fructose, mainly consumed as high-fructose corn syrup (HFCS), remain high [1]. Glucose and fructose induce the production of advanced glycation end-products (AGEs) in the human body, and toxic and non-toxic AGEs have been detected among the various types of AGE structures generated in vivo [2,3]. We previously detected AGEs derived from glyceraldehyde (GA), a glucose/fructose metabolism intermediate, which are also known as GA-AGEs. We designated GA-AGEs as toxic AGEs (TAGE) because of their cytotoxicity and involvement in lifestyle-related diseases (LSRD), such as non-alcoholic fatty liver disease (NAFLD), which ranges from simple steatosis to non-alcoholic steatohepatitis (NASH) [2,3]. We proposed that (i) GA induces the generation of TAGE in intracellular components; (ii) TAGE accumulates in cells and causes cell damage; and (iii) TAGE leak into blood [2,3,4]. We already revealed the intracellular generation of TAGE in the livers of both simple steatosis and NASH patients [5], and previously investigated serum TAGE levels in healthy humans [6,7,8] and patients with LSRD, such as NASH [5,9,10,11].

Although circulating TAGE have been detected in the blood of healthy humans, it currently remains unclear whether intracellular TAGE are present in normal organs, such as the liver. We speculated that intracellular TAGE are generated in the normal liver. To prove our hypothesis, we maintained male Wister/ST rats (11 weeks) for 13 weeks with the intake of 10% HFCS 55 (HFCS beverage) or 12 weeks with a *Lactobacillus* beverage. The HFCS beverage was prepared to simulate HFCS-containing beverages that are commercially available, and the *Lactobacillus* beverage is a commercial drink. Serum TAGE levels and intracellular TAGE levels were measured at the end point of this examination. To assess steatosis, inflammation, and fibrosis in the liver of rats that drank the HFCS beverage and *Lactobacillus* beverage, hematoxylin and eosin (H.E.) staining, Oil red staining, and Sirius red staining were performed. The expression of type IV collagen, SIRT1, and heat shock protein 70 (HSP70), proteins that are associated with steatosis, inflammation, and fibrosis, was analyzed in the livers of rats that drank the HFCS beverage using Western blotting (WB). 

## 2. Materials and Methods

### 2.1. Animals

All experiments using rats were approved by the Committee on Experimental Animals at Kanazawa Medical University and conducted in accordance with their guidelines. Male Wister/ST rats (10 weeks) were obtained from Sankyo Lab Service Co., Inc. (Tokyo, Japan). 

### 2.2. Reagents and Antibodies

GA was purchased from Nacalai Tesque, Inc. (Kyoto, Japan); and 3-[(3-Cholamido-propyl)-dimethyl-ammonio]-1-propane sulfonate) (CHAPS) and 3,3′-diaminoenzidine tetrahydrochloride (DAB) were obtained from Dojindo Laboratories (Kumamoto, Japan). An ethylenediamine-*N*,*N*,*N*′,*N*′-tetraacetic acid (EDTA)-free protease inhibitor cocktail was obtained from Roche Applied Science (Penzberg, Germany). The protein assay kit for the BCA method was purchased from Thermo Fisher Scientific Inc. (Waltham, MA, USA). The protein assay kit for the Bradford method was obtained from Takara Bio, Inc. (Otsu, Japan). CE-2, which was produced in 2015 (normal diet; 8.89% moisture, 24.88% crude protein, 5.03% crude fat, 4.63% crude fiber, 6.79% crude ash, and 49.78% nitrogen-free extracts (NFE). Also, 343.9 kcal/100 g) was obtained from Clea Japan Inc. (Tokyo, Japan). HFCS (HFCS 55; 41.25 g fructose, and 33.75 g glucose, 25.00 g moisture per 100 g. Also, 1.37 g/mL, 276 kcal/100 g) was obtained as a commercial sweetener, and followed the Japanese Agricultural Standard. A Sprague Dawley (SD) rat primary hepatocyte kit was purchased from Cosmo Bio Co., Ltd. (Tokyo, Japan). Fructose, 5.6 mM glucose Dulbecco’s modified Eagle’s medium (D-MEM), and penicillin/streptomycin were obtained from Sigma-Aldrich Co. LLC. (St. Louis, MO, USA). Fetal bovine serum albumin was obtained from Bovogen-Biologicals (VIC, Australia). CE-2, which was produced in 2016 (a normal diet; 8.91% moisture, 25.33% crude protein, 4.38% crude fat, 4.88% crude fiber, 6.83% crude ash, and 49.67% NFE. Also, 339.4 kcal/100 g) was obtained from Clea Japan Inc. (Tokyo, Japan). The *Lactobacillus* beverage (1.23% crude protein, 0.15% crude fat, and 17.69% crude carbohydrate (5.08% fructose, 5.38% glucose, 4.15% sucrose, 1.54% lactose, and 1.54% other carbohydrates.) Also, 50 kcal/65 mL) was purchased as a commercial drink. The Western re-probe kit was purchased from Funakoshi Co., Ltd. (Tokyo, Japan). A horseradish peroxidase (HRP)-linked molecular marker was obtained from Bionexus (Oakland, CA, USA). A mouse-monoclonal anti-SIRT1 antibody (anti-SIRT1 antibody) was purchased from Novus Biologicals (Centennial, CO, USA). A rabbit polyclonal anti-HSP70 antibody (anti-HSP70 antibody) and mouse monoclonal anti-β-actin antibody (anti-β-actin antibody) were purchased from Abcam (Cambridge, United Kingdom). An HRP-linked goat anti-rabbit IgG antibody was purchased from DAKO (Glostrup, Denmark), and HRP-linked goat anti-mouse and HRP-linked donkey anti-rabbit IgG antibodies were obtained from Thermo Fisher Scientific Inc. Somnopentyl (Pentobarbital sodium) was purchased from Kyoritsuseiyaku Co. (Tokyo, Japan). Malinol was purchased from Muto Pure Chemical Co., Ltd. (Tokyo, Japan) and Multi Mount 480 was obtained from Matsunami Glass Ind., Ltd. (Osaka, Japan). The Picro-Sirius red stain kit was obtained from Cosmo Bio Co., Ltd. (Tokyo, Japan). TAGE-bovine serum albumin (TAGE-BSA) and the rabbit polyclonal anti-TAGE antibody (anti-TAGE antibody) were prepared as described previously [12]. All other reagents and kits not indicated were purchased from Wako Pure Chemical Industries Ltd. (Osaka, Japan). The adipose tissue of Oil Red O staining for positive control was obtained from Histology Laboratory of Research Support Center of Medical Research Institute of Kanazawa Medical University.

### 2.3. Animal Breeding and Experimental Design

Rats were allocated a dietary regimen of drinking water and a normal diet (CE-2 produced in 2015, Clea Japan Inc.) for an initial period of 1 week. They were then divided into two groups, with each group comprising five rats. Group 1 served as the control. Group 2 drank 10% HFCS 55 (HFCS beverage) (HFCS group). Each group of rats was maintained for 13 weeks. They were housed in standard polypropylene cages (one rat/cage) and maintained under controlled room temperature (r.t., 24 ± 3 °C) and humidity (50 ± 20%) with a 12-h light/dark cycle [13]. The body weights of all animals were measured, and they were then euthanized by barbital overdose (intravenous injection, 150 mg/kg pentobarbital sodium) to collect blood and tissue under non-fasted conditions. Blood was collected for serum separation. Hepatic tissues were excised and washed with ice-cold phosphate-buffered saline without Ca^++^ and Mg^++^ ((PBS(-)). After hepatic tissues had been blotted with filter paper to remove PBS (-), liver weights were measured. Hepatic tissues were cut into blocks, with one block then being placed into a tube (Sumitomo Bakelite Co., Ltd., Tokyo, Japan), immersed in liquid nitrogen for a few minutes, and preserved at −80 °C. Other blocks were fixed in 4% formaldehyde, embedded in paraffin, and sectioned. Rats were also allocated to a dietary regimen of drinking water and the normal diet (CE-2 produced in 2016) for an initial period of 1 week. They were then divided into two groups, with each group comprising five rats. Group 1 served as the control. Group 2 drank the *Lactobacillus* beverage. Each group of rats was maintained for 12 weeks.

### 2.4. Biochemical Analysis of Blood

#### 2.4.1. Serum TAGE Levels

Briefly, each well of a 96-well microtiter plate was coated with 1.0 μg/mL TAGE-BSA and incubated overnight in a cold room. Wells were washed three times, with 0.3 mL of PBS (-) containing 0.05% Tween 20 (PBS-T). Wells were then blocked by incubation for 1 h with 0.2 mL of a solution of PBS (-) containing 1% BSA. After washing with PBS-T, test samples (50 µL) were added to each well as a competitor for 50 µL of the anti-TAGE antibody (1:1000), followed by incubation at r.t. for 2 h with gentle shaking on a horizontal rotary shaker. Wells then were washed with PBS-T and developed with alkaline phosphatase-linked anti-rabbit IgG utilizing p-nitrophenyl phosphate as the colorimetric substrate. Results were expressed as TAGE units (U) per milliliter of serum, with 1 U corresponding to 1.0 μg of a TAGE-BSA standard as described previously [12]. Sensitivity and intra- and interassay coefficients of variation were 0.01 U/mL and 6.2% and 8.8%, respectively [14].

#### 2.4.2. Parameters Other than Serum TAGE Levels

Measurements of glucose, glycoalbumin, blood urea nitrogen (BUN), uric acid (UA), Ca, aspartate aminotransferase (AST), alanine aminotransferase (ALT), lactate dehydrogenase (LDH), creatinine kinase (CK), total cholesterol (T-CHO), triglycerides (TG), low-density lipoprotein cholesterol (LDL-C), and total bilirubin (T-BIL) were outsourced (Oriental Yeast Co., Ltd., Tokyo, Japan).

### 2.5. H.E. Staining of Hepatic Sections

After the removal of paraffin with xylene, ethanol, ethanol/water (90%, 80%, and 70% ethanol), and water, tissue sections were counterstained briefly with H.E. After the removal of reagents with ethanol/water (70%, 80%, and 90% ethanol), ethanol, and xylene, sections were coated with malinol and then observed under the NanoZoomer slide scanner (Hamamatsu Photonics K.K., Hamamatsu, Japan).

### 2.6. Oil Red O Staining of Hepatic Sections

Six point one mM Oil Red O was dissolved in isopropanol, and diluted with water to prepare 3.7 mM Oil Red O solution. Tissue sections were washed with water. They were then incubated in Oil Red reagents at r.t for 10 min, followed by the removal of Oil Red O solution with water. They were incubated in hematoxylin at r.t. for 5 min and washed with water. Sections were coated with malinol and then observed under the NanoZoomer slide scanner (Hamamatsu Photonics K.K.).

### 2.7. Sirius Red Staining of Hepatic Sections

After the removal of paraffin with xylene, ethanol, ethanol/water (90%, 80%, and 70% ethanol), and water, tissue sections were incubated in Sirius Red reagent at r.t. for 1 h. After the removal of the reagent with 0.5% acetic acid and ethanol, sections were coated with Multi Mount 480 and then observed under the NanoZoomer slide scanner (Hamamatsu Photonics K.K.).

### 2.8. Preparation of Hepatic Tissue Homogenates

#### 2.8.1. Samples for the WB Analysis

Blocks of hepatic tissues were cut and washed with ice-cold PBS(-), which was then removed using filter paper. Blocks were then placed into tubes for homogenization (WATSON Co., Ltd., Tokyo, Japan) using two zirconia beads with a diameter of 5.0 mm (AS ONE Co., Ltd., Osaka, Japan). After being immersed in liquid nitrogen for 30 s followed by the addition of a buffer [radioimmunoprecipitation assay buffer (Thermo Fisher Scientific Inc.) and solution of the EDTA-free protease inhibitor cocktail (9:1)], blocks were homogenized using BEADS CRUSHER μT-01 (TAITEC Co., Ltd., Saitama, Japan). Tissue extracts were then incubated on ice for 20 min, centrifuged at 10,000× *g* and 4 °C for 15 min, and supernatants were collected as tissue lysates.

#### 2.8.2. Samples for the Slot Blotting (SB) Analysis

Tissue blocks were washed with ice-cold PBS(-) and immersed in liquid nitrogen for 30 s Blocks were placed in tubes to which the buffer [a solution of 2M thiourea, 7 M urea, 4% CHAPS, and 30 mM Tris (hydroxymethyl) aminoethane (Tris), and a solution of the EDTA-free protease inhibitor cocktail (9:1)] was added [15]. Methods for the homogenization and collection of tissue lysates were described in the method for the WB analysis.

### 2.9. WB Analysis of Tissue Lysates

Protein concentrations were assessed using the protein assay kit for the BCA method with BSA as a standard. Tissue lysates (15 μg of protein) were mixed with sodium dodecyl sulphate (SDS) sample buffer and 2-mercaptoethanol (Sigma-Aldrich, St. Louis, MO, USA), and then heated at 95 °C for 5 min. Equal amounts of cell extracts were resolved by 4–15% gradient SDS-polyacrylamide gel (Bio-Rad, Hercules, CA, USA) electrophoresis and transferred onto polyvinylidene difluoride (PVDF) membranes (0.45 μm; Millipore, MA, USA). Membranes were blocked at r.t. for 30 min using 5% skimmed milk in PBS-T (SM-PBS-T). We then used 0.5% SM-PBS-T for washing or as the solvent of antibodies. After washing twice, membranes were incubated with the anti-type IV collagen antibody (1:1000; ab6586), anti-SIRT1 antibody (1:2000; NBP1-51641), and anti-HSP70 antibody (1:8000; ab94368). PVDF membranes were washed four times with 0.5% SM-PBS-T and incubated with a secondary antibody at r.t. for 1 h. The secondary antibodies used were as follows: The HRP-linked goat anti-mouse IgG antibody (1:5000; Product No. 31432) or HRP-linked donkey anti-rabbit IgG antibody (1:2000; Product No. 31458). Membranes were then washed with PBS-T. Immunoreactive complexes were visualized using the ImmunoStar LD kit. Band densities on the membranes were measured using the LAS-4000 fluorescence imager (GE Healthcare, Tokyo, Japan) or Fusion FX fluorescence imager (M&S Instruments Inc., Osaka, Japan), and expressed in arbitrary units (AU). Equivalent sample loading was confirmed by stripping membranes with the Western re-probe kit, and this was followed by blotting with the anti-β-actin antibody (1:3000; ab3280). 

### 2.10. SB Analysis of Tissue Lysates

This analysis was performed as previously described with some modifications [15]. Protein concentrations were measured by the protein assay kit for the Bradford method using BSA as the standard. In the detection of TAGE, equal amounts of tissue lysates (10 μg of protein), the HRP-linked molecular marker, and TAGE-BSA were loaded onto PVDF membranes fixed in the SB apparatus (Bio-Rad). PVDF membranes were then cut to prepare two membranes, and were blocked at r.t. for 1 h using 5% SM-PBS-T. After this step, we used 0.5% of SM-PBS-T for washing or as the solvent of antibodies. After washing twice, membranes were incubated with (1) the anti-TAGE-antibody (1:1000) or (2) neutralized anti-TAGE-antibody (a mixture of the anti-TAGE-antibody (1:1000) and 250 µg/mL of TAGE-BSA) at 4 °C overnight. Membranes were then washed four times. Proteins on the membrane were incubated with the HRP-linked goat anti-rabbit IgG antibody (1:2000, REF0448) at r.t. for 1 h. After washing three times with PBS-T, membranes were moved into PBS. Immunoreactive proteins were detected with the ImmunoStar LD kit, and band densities on the membranes were measured using the Fusion FX fluorescence imager (M&S Instruments Inc., Osaka, Japan). The densities of HRP-linked molecular marker bands were used to correct for differences in densities between membranes. The amount of TAGE in tissue lysates was calculated based on a calibration curve for TAGE-BSA.

### 2.11. Immunostaining of Hepatic Sections with the Anti-TAGE Antibody

After the removal of paraffin with xylene, ethanol, and ethanol/water (90%, 80%, and 70% ethanol), and water, tissue sections were treated with 3% hydrogen peroxide (H_2_O_2_) in PBS(-)for 10 min to inhibit endogenous peroxidase and then treated with 3% BSA PBS(-) for 1 h for blocking. After this step, we used 1% BSA PBS(-)as the solvent of antibodies and 0.1% BSA PBS(-)for washing. After being washed, tissue sections were incubated with the anti-TAGE antibody (1:50) for 1 h, washed three times, and then incubated with the HRP-linked goat anti-rabbit IgG antibody (1:100, REF0448) for 1 h. Tissue sections were washed three times, with 0.1% BSA PBS(-), and washed once with PBS(-). Tissue sections were incubated with 0.02% DAB and 0.015% H_2_O_2_ in PBS(-) for 5 min. Tissue sections were treated with ethanol/water (70%, 80%, and 90% ethanol), ethanol, and xylene to remove water. They were then coated with Multi Mount 480. Samples were observed under the NanoZoomer slide scanner (Hamamatsu Photonics K.K.).

### 2.12. Preparation Medium to Incubate SD Rat Primary Hepatocytes

To obtain control medium or high fructose medium, 5.6 mM glucose D-MEM was added to 25 mM mannitol or 25 mM fructose, respectively.

### 2.13. Cell Culture

Hepatocytes obtained from male SD rats (5 weeks) were cultured in a 25-cm^2^ flask (1.0 × 10^5^ cells/cm^2^, *n* = 8) in the SD rat primary hepatocyte kit. The medium from the kit was removed, and cells were washed with PBS(-) and then incubated in control (*n* = 4) or high fructose medium (*n* = 4). Medium changes were performed every 24 h and cells were cultured for 120 h.

### 2.14. SB Analysis of Cell Lysates

SD rat primary hepatocyte cells were washed with PBS(-), and lysed in buffer [a solution of 2 M thiourea, 7 M urea, 4% CHAPS, and 30 mM Tris, and a solution of the EDTA-free protease inhibitor cocktail (9:1)] [15] After this step, the SB analysis of cell lysates (4.0 μg of protein) was performed as described in “*SB analysis of tissue lysates*”. 

### 2.15. Statistical Analysis 

Stat Flex (ver. 6) software (Artech Co., Ltd., Osaka, Japan) was used for statistical analyses. Data are expressed as means ± S.E. or means ± S.D. When statistical analyses were performed on data, the Mann-Whitney U-test was used for an analysis of variance. *p*-Values < 0.05 were considered to be significant. 

## 3. Results

### 3.1. Intake Parameters for the Normal Diet and HFCS Beverage and Body and Liver Weights

The control and HFCS groups ate 16.9 and 12.0 g of the normal diet each day (Table 1). The HFCS group ate approximately 0.7-fold the amount of crude protein and crude fat consumed by the control group. The HFCS group drank 127.4 mL of the HFCS beverage each day, which contained 7.2 g of fructose and 5.9 g of glucose, and energy consumed per day by the HFCS group was 1.5-fold that of the control group. No significant differences were observed in body or liver weights between the HFCS and control groups. The liver index in the HFCS group was 108% that in the control group (Table 1).

### 3.2. Serum and Plasma Biochemistries in the HFCS Group

Serum TAGE levels in the control and HFCS groups were 10.08 and 14.58 U/mL, respectively, and were 145% higher in the HFCS group than in the control group (Table 2). Glucose levels were 14.08 mM in the HFCS group and 9.70 mM in the control group. TG levels were 1.61 mM in the HFCS group and 0.79 mM in the control group. BUN and ALT levels in the HFCS group were 48% and 57% those in the control group (Table 2). 

### 3.3. Histological Analysis of the Liver with H.E., Oil Red O, and Sirius Red Staining

Each score for steatosis, inflammation, and ballooning in the control and HFCS groups was analyzed with H.E. staining (Figure 1 and Table 3). The NAFLD activity score (NAS) was 0 against steatosis, inflammation, and ballooning in the control and HFCS groups. Therefore, H.E. staining showed no significant differences in the liver between the HFCS and control groups. Oil Red O and Sirius Red staining showed no significant differences in the area of fat, muscle fibers, or collagen in the liver between the HFCS and control groups (Figure 2 and Figure 3).

### 3.4. Expression Partterns of Proteins Associated with Steatosis, Inflammation, and Fibrosis

No significant differences were observed in the expression of type IV collagen, SIRT1, or HSP70 in the liver between the HFCS and control groups (Figure 4 and Appendix A). 

### 3.5. Calculation of Intracellular TAGE Levels in the Liver and Its Relationship with Serum TAGE Levels in the HFCS Group

Intracellular TAGE levels in the liver were 0.84 and 2.88 μg/mg protein in the control and HFCS groups, respectively (Figure 5a). Intracellular TAGE levels were more than 3-fold higher in the HFCS group than in the control group. Intracellular TAGE levels in the liver positively correlated with serum TAGE levels, and the correlation coefficient was 0.911 (Figure 5b).

### 3.6. Histological Analysis of the Liver with Anti-TAGE Antibody Staining

Immunostaining with the anti-TAGE antibody revealed the intracellular generation of TAGE in the liver in the control and HFCS groups. Intracellular TAGE levels in the liver were higher in the HFCS group than in the control group (Figure 6).

### 3.7. Calculation of Intracellular TAGE Levels in SD Rat Primary Hepatocytes

Intracellular TAGE levels in SD rat primary hepatocytes incubated in control and high fructose media were 0.47 and 6.13 μg/mg protein, respectively (Figure 7). These cells did not exhibit cell death.

### 3.8. Intake of the Normal Diet and Lactobacillus Beverage and Body and Liver Weights

The control and *Lactobacillus* beverage groups ate 19.9 and 10.3 g of the normal diet each day (Table 4). The *Lactobacillus* beverage group drank 54.8 mL of the *Lactobacillus* beverage each day, which contained 2.8 g of fructose, 3.0 g of glucose, and 2.3 g of sucrose. The *Lactobacillus* beverage group ate approximately 0.5-fold the amounts of crude protein and crude fat consumed by the control group. No significant differences were observed in body weight, liver weight, or the liver index between the *Lactobacillus* beverage and control groups.

### 3.9. Serum and Plasma Biochemistries in the Lactobacillus Beverage Group 

Serum TAGE levels were 10.12 and 10.02 U/mL in the control and *Lactobacillus* beverage groups. Glucose levels were 14.65 mM in the *Lactobacillus* beverage group and 9.73 mM in the control group. BUN levels in the *Lactobacillus* beverage group were 65% of those in the control group (Table 5). 

### 3.10. Histological Analysis of the Liver with H.E., Oil Red O, and SIRIUS Red Staining in the Lactobacillus Beverage Group

Each score for steatosis, inflammation, and ballooning in the control and *Lactobacillus* groups was analyzed with H.E. staining (Figure 8 and Table 6). NAS was 0 for steatosis, inflammation, and ballooning in the control and *Lactobacillus* groups. H.E. staining showed no significant differences in the liver between the *Lactobacillus* and control groups. Oil Red O staining and Sirius Red staining indicated no significant differences in the fat area in the liver between the HFCS and control groups (Appendix A). 

### 3.11. Calculation of Intracellular TAGE Levels in the Lactobacillus Beverage Group

Intracellular TAGE levels in the liver were 0.65 and 0.67 μg/mg protein in the control and *Lactobacillus* beverage groups, respectively (Figure 9). No significant differences were observed in intracellular TAGE levels between the control and *Lactobacillus* beverage groups.

### 3.12. Histological Analysis of the Liver with Anti-TAGE Antibody Staining in the Lactobacillus Group

Immunostaining with the anti-TAGE antibody revealed the intracellular generation of TAGE in the liver in both the control and *Lactobacillus* groups. No significant differences were observed in intracellular TAGE levels between the control and *Lactobacillus* beverage groups (Appendix A).

## 4. Discussion

HFCS contains more fructose than glucose, and both sugars are present as monosaccharides. This “free fructose” is preferred by beverage and food manufacturers because it significantly increases the perception of sweetness [16]. Industrially, HFCS is frequently found in beverages and pre-packaged foods. The most common form of HFCS is HFCS 55 (fructose:glucose = 55:45) [17]. We previously reported the amounts of total sugar and free glucose in typical beverages in Japan and also calculated fructose plus sucrose; approximately 40% of the beverages tested contained 25 g or more sugar per bottle in a standard serving size [3]. Based on these findings, we prepared an HFCS beverage for the present study. We maintained male Wister/ST rats (11 weeks) for 13 weeks and provided an HFCS beverage, of which the HFCS group drank 127.4 mL each day and gained 7.2 g of fructose and 5.9 g of glucose (Table 1). NFE mainly consists of carbohydrates. Therefore, the HFCS group gained more than 7.2 g of fructose and 5.9 g of glucose each day, indicating that the control and HFCS groups gained approximately 8.4 and 19.1 g of carbohydrate each day. Although the HFCS group only ate 0.7-fold the amounts of protein and fat consumed by the control group, it gained 1.5-fold the amount of energy of the control group each day. These results indicate that rats that consumed the HFCS beverage avoided eating the normal diet; however, the excess amounts of fructose and glucose in the beverage supplied sufficient energy to rats.

Body weight, liver weight, and the liver index under fasted conditions were previously reported to have significantly increased in NAFLD model rats in the NAFLD stage [18,19]. In the present study, no significant differences were observed in body or liver weights between the HFCS and control groups, and only the liver index in the HFCS group under non-fasted conditions increased to 108% that in the control group. The liver index of the NAFLD stage of NAFLD model rats under fasted conditions increased to approximately 140% that of the control [18,19]. We considered livers in the HFCS group to be at an increased risk of NAFLD even though they were not at the NAFLD stage, and this result may be due to the increased amount of fructose/glucose.

To analyze serum and plasma biochemistries, we collected blood from non-fasted rats. Although some parameters, such as plasma glucose and TG, may have been higher than those in fasted rats [20,21], we considered the stress experienced by rats subjected to this method to be less than that by fasted rats. On the other hand, serum TAGE levels, which we regarded as the most important biomarker, were not affected by the fasted/non-fasted treatment. AGEs containing TAGE are generated by the Maillard reaction. This reaction begins with the conversion of reversible Schiff base adducts to more stable, covalently bound Amadori rearrangement products. Over the course of days to weeks, these Amadori products undergo further rearrangement reactions to generate irreversibly bound moieties known as AGEs [3]. The results of biochemical analyses of serum and plasma revealed that TAGE, glucose, TG, BUN, and ALT levels were significantly higher/lower in the HFCS group than in the control group (for more detailed information, refer to Table 2). These results may be attributed to TAGE serum levels being 145% higher in the HFCS group than in the control group. This indicates the generation of intracellular TAGE in organs because we previously speculated that TAGE in blood leaked from some organs [3]. Although we have already detected intracellular TAGE in human cells (hepatocytes [22,23,24], pancreatocytes [15], and neuroblasts [25]), and rat primary cardiomyocytes [26] treated with GA *in vitro*, we consider the liver to be the main organ generating TAGE because it has two active GA-forming pathways: (i) The glycolytic pathway and (ii) fructose metabolic pathway, in humans [2,3] and rats [27,28], and GA induces the generation of TAGE. Furthermore, we previously reported that the viability of the human hepatocyte cell lines Hep3B and HepG2 decreased in a manner that depended on the generation of intracellular TAGE [22,24]. Therefore, intracellular TAGE may be secreted or released from cells. We previously indicated the usefulness of serum TAGE levels as a biomarker for the prevention/early diagnosis of LSRD and evaluation of the efficacy of treatments [3]. We demonstrated that serum TAGE levels were significantly higher in NASH patients than in patients with simple steatosis and healthy humans [5]. These findings also revealed that TAGE serum levels were higher in NASH without non-B or non-C hepatocellular carcinoma (NBNC-HCC) patients than in healthy humans, but were higher in NBNC-HCC patients than in those with NASH without NBNC-HCC [9]. Moreover, these levels decreased when treatments for NASH patients with dyslipidemia improved the pathology of the liver [10]. Therefore, elevated serum TAGE levels indicate the increased intracellular generation of TAGE in the rat liver. 

Plasma glucose and TG levels were 145% and 200% higher, respectively, in the HFCS group than in the control group in the present study (Table 2). Kaji et al. reported non-fasted plasma glucose and TG levels in male WBN/Kob rats (11 weeks) and age-matched male WBN/Kob-*Lepr^fa^* rats, which are a type 2 diabetes mellitus (T2DM) model in the pre-diabetic phase [29]. Plasma glucose levels in WBN/Kob rats and WBN/Kob-*Lepr^fa^* rats were approximately 6.7 and 10 mM, respectively. Although plasma glucose levels were approximately 150% higher in WBN/Kob-*Lepr^fa^* rats than in WBN/Kob rats, they did not develop T2DM. Furthermore, non-fasted plasma TG levels in WBN/Kob rats and WBN/Kob-*Lepr^fa^* rats were approximately 1.3 and 5.8 mM, respectively. There was no description regarding whether WBN/Kob-*Lepr^fa^* rats developed dyslipidemia. In another study, male Wister rats (7 weeks) and age-matched male WBN/Kob-*Lepr^fa^* rats were bred and fed standard rat chow for four weeks, and their fasted plasma TG levels were approximately 0.6 and 4.5 mM, respectively [30]. These WBN/Kob-*Lepr^fa^* rats developed dyslipidemia. Based on these findings, the HFCS group was not expected to develop T2DM or dyslipidemia. Although BUN in the HFCS group decreased to 48% that in the control group, a lack of protein in the normal diet may have induced this decrease [31]. A previous study reported that ALT activity was enhanced by vitamin B_6_ [32]. ALT activity may have been reduced by the lack of vitamin B_6_ in the normal diet, and, thus, was decreased in the biochemical analysis of serum. 

Steatosis is detected at an early stage, whereas inflammation and fibrosis do not appear in the liver. At the NASH stage, steatosis, inflammation, and fibrosis are detected in the liver [18,19,33].

H.E. staining did not reveal any significant differences in the liver between the HFCS and control groups. Steatosis, inflammation, and fibrosis were not detected (Figure 1). Although the method for H.E. staining is simple, Oil Red O staining or Sirius Red staining may be used to analyze the presence of fat or muscle fibers/collagens, which is associated with steatosis or fibrosis in the liver [19,34,35,36]. Therefore, we performed Oil red O staining and Sirius red staining after the H.E. staining analysis. No significant difference was observed in fat in the liver between the control and HFCS groups with Oil Red O staining (Figure 2). This result indicates that fat accumulation was not enough to be the NAFLD stage in the liver in the HFCS group. In the natural course of chronic liver disease, remodeling of the extracellular matrix (ECM) leads to progressive fibrosis [37]. During liver fibrosis, activated hepatic stellate cells transdifferentiate to myofibroblast-like secreting profibrogenic factors and excessive ECM proteins. These ECM proteins include type I, III, and IV collagens. Type IV collagen is the main type of collagen in the basement membrane, increasing by up 14-fold during cirrhosis [37], and serum type IV collagen is a useful biomarker of liver fibrosis in NAFLD [38]. Sirius Red staining showed that the areas of muscle fibers and type I or III collagen were similar in the HFCS and control groups (Figure 3). Since this assessment was only based on a visual examination, we examined the expression of type IV collagen using the WB analysis. The expression of type IV collagen in the liver was not significantly different between the HFCS and control groups (Figure 4a and Appendix A). We then attempted to analyze other proteins associated with steatosis, inflammation and fibrosis in the rat liver [36]. We previously reported that the human hepatocyte cell line, Hep3B, which generated intracellular TAGE, showed elevated mRNA levels of the C-reactive protein (CRP), and this indicated that intracellular TAGE induce inflammation in the liver [20]. However, SIRT1 was recently targeted by anti-steatosis and anti-inflammation treatments [39], and HSP70 has been implicated in both inflammation and fibrosis [40]. Although SIRT1 and HSP70 levels were previously reported to be decreased in rat livers with steatosis, inflammation, and fibrosis [39,40], SIRT1 and HSP70 levels were not lower in the HFCS group than in the control group (Figure 4b,c,e,f, Appendix A). Sirius Red staining and the WB analysis did not detect steatosis, inflammation, or fibrosis. These results also indicate that NAFLD had not yet developed.

The results of the SB analysis and immunostaining of hepatic tissues suggested that the intracellular generation of TAGE in the liver was greater in the HFCS group than in the control group (Figure 5a and Figure 6). Dietary fructose may generate other AGEs [41,42]. Although we predicted a high amount of intracellular TAGE to be generated by HFCS, the level of intracellular TAGE in the liver in the HFCS group was 2.88 μg/mg protein. Therefore, we speculate that fructose generates high levels of intracellular TAGE in the liver. Previous studies reported that SD rats that ate a high fructose diet developed metabolic syndrome [43,44]. Therefore, we analyzed the quantity of intracellular TAGE generated in SD rat primary hepatocytes by 25 mM fructose. The fructose concentration used was based on a previous study by Patel et al. [45]. They fed F344 rats a high fructose diet, and serum fructose was measured. Other F344 rats were then perfused with 50 mM fructose Ringer solution (intestinal perfusion), and serum fructose was measured. Based on the findings obtained, we selected 25 mM fructose as the physiological concentration to treat SD rat primary hepatocytes. SD rat primary hepatocytes were incubated in 5.6 mM glucose D-MEM without fructose (control medium) or 5.6 mM glucose and 25 mM fructose D-MEM (high fructose medium) for 120 h. Intracellular TAGE levels were 0.47 and 6.13 μg/mg protein in hepatocytes incubated in control and high fructose media, respectively (Figure 7). These results indicate that fructose generates TAGE in rat hepatocytes. 

We proposed some hypotheses for the mechanisms by which intracellular TAGE in the liver contributes to the progression of inflammation [22] and fibrosis [24] in NASH. Hypothesis 1: Intracellular TAGE in hepatocytes up-regulate the expression of CRP to induce inflammation. Hypothesis 2: (i) Intracellular TAGE in hepatocytes induce cell damage and necrotic cell death. (ii) TAGE leak from hepatocytes and combine with the receptors for AGEs on hepatic stellate cells. (iii) Hepatic stellate cells induce and promote fibrosis. However, the generation of 2.88 μg/mg protein of TAGE in the livers of Wister/ST rats may be insufficient to cause cell damage or induce inflammation and fibrosis (Figure 1, Figure 2, Figure 3 and Figure 4 and Figure 5a). In primary cardiomyocytes in neonatal Wister/ST rats, 12.0 μg/mg protein of TAGE completely stopped beating and decreased cell viability to 39% [26]. In primary hepatocytes in SD rats, cells generated 6.13 μg/mg protein of TAGE and cell death was not detected. Although the relationship between the generation of intracellular TAGE and cytotoxicity against primary hepatocytes in Wister/ST rat currently remains unclear, we proposed that more than 6.13 μg/mg protein of TAGE in the liver is needed to induce prominent cell damage/cell death and cause TAGE leakage from hepatocytes. If the concentration of HFCS in the HFCS beverage was more than 10% or the period of consumption was longer than 13 weeks, intracellular TAGE levels in the liver of the HFCS group may have increased to that needed to induce cytotoxicity. Furthermore, we need to consider the reason why fat did not accumulate in the liver in the HFCS group because simple steatosis was not induced in this study. The accumulation of fat in the liver is affected by dietary saccharides and fats [36], and the reason for the lack of a significant difference between the control and HFCS groups may be attributed to the two different intake routes: (i) Dietary saccharides (particularly fructose) are transported into hepatocytes, and free fatty acids are generated; and (ii) dietary fats are transported into adipose tissue, and blood fatty acids are generated and transported into hepatocytes [36]. In the present study, fructose/glucose in the HFCS beverage and saccharides in the normal diet are needed for the generation of TAGE or fatty acids; however, we were unable to clarify the ratio required. We calculated the amount of dietary fat in the control and HFCS groups (Table 1). The HFCS group ate approximately 0.7-fold the dietary fat consumed by the control group. The reason that fat accumulation was not detected, and, thus, steatosis was not induced in the liver in the HFCS group may be due to the lack of dietary fat in the normal diet. If a large amount of fat is generated in the liver, and simple steatosis occurs, intracellular TAGE may promote inflammation and fibrosis to induce the NASH stage.

We investigated whether the *Lactobacillus* beverage, a commercial drink that contains saccharides, such as fructose, glucose, and sucrose, generates intracellular TAGE in the rat liver and induces the NAFLD phenotype. The *Lactobacillus* beverage group drank 54.8 mL of the beverage, which contained 2.8 g of fructose, 3.0 g of glucose, and 2.3 g of sucrose, and ate 10.3 g of the normal diet each day (Table 4). Since sucrose is metabolized to fructose and glucose in the body, the *Lactobacillus* beverage group gained approximately 4.0 g of fructose and 4.2 g of glucose each day from the *Lactobacillus* beverage, with the intake of fructose and glucose is approximately 56% and 71% those in rats that consumed the HFCS beverage. The increase in carbohydrates in the *Lactobacillus* beverage group was approximately 14.8 g because NFE mainly consists of carbohydrates. This value was 77% in the HFCS group. However, no significant difference was observed in the liver index between the control and *Lactobacillus* beverage groups (Table 4); only plasma glucose and BUN were altered in the *Lactobacillus* beverage group (Table 5). The HFCS beverage induced an increase in the liver index and decreased ALT and TG (Table 1 and Table 2), whereas the *Lactobacillus* beverage did not (Table 4 and Table 5). One of the reasons for this may be the differences in their components. (i) The amounts of fructose and glucose were higher in the HFCS beverage than in the *Lactobacillus* beverage. (ii) The *Lactobacillus* beverage contains various components other than protein, fat, and carbohydrates. (iii) The *Lactobacillus* beverage contains *Lactobacillus*, which is alive.

On the other hand, steatosis, inflammation, and fibrosis in the liver were not detected in the *Lactobacillus* beverage group (Figure 8, Table 6, Appendix A) or HFCS group (Figure 1, Figure 2 and Figure 3). Therefore, the *Lactobacillus* beverage did not promote the generation of intracellular TAGE or increase serum levels of TAGE (Figure 9 and Appendix A, Table 5) over those in the HFCS group (Figure 5a and Figure 6, Table 2). Since we hypothesized that intracellular TAGE induce inflammation and fibrosis, the consumption of 54.8 mL of the *Lactobacillus* beverage by rats each day may not have been sufficient to increase the generation of intracellular TAGE in the liver or promote the leakage of TAGE from hepatocytes into the vascular system (Figure 9 and Table 4 and Table 5). The generation of intracellular TAGE may compete against the generation of other AGEs that are metabolized from fructose/glucose. Therefore, the accelerated generation of intracellular TAGE may require the intake of large amounts of fructose and glucose. If other components in the *Lactobacillus* beverage or *Lactobacillus* inhibit the generation or promote the degradation of intracellular TAGE, this will be a novel effect of the *Lactobacillus* beverage.

The reason for the absence of fat accumulation and steatosis in the liver in the *Lactobacillus* beverage group may be due to the lack of dietary fat. The amount of dietary fat consumed by the *Lactobacillus* beverage group was approximately 0.6-fold that by the control group (Table 4).

Based on the results obtained in the HFCS group, a lifestyle in which sweet beverages that contain large amounts of fructose/glucose, such as HFCS, are drunk may increase the risk of NAFLD via the excess generation of intracellular TAGE in the liver. A survey on NAFLD patients revealed that the dietary intake of sweet beverages was 5-fold higher in these patients than in healthy humans [46].

The correlation coefficient of intracellular TAGE in the liver and serum TAGE levels was 0.911 (*p* < 0.001) (Figure 5b), which was very high. This result indicated that serum TAGE levels have potential as a useful biomarker to predict the intracellular generation/accumulation of TAGE in the liver.

## 5. Conclusions

We demonstrated that the excess consumption of HFCS may decrease the intake of a normal diet containing multiple nutrients, such as protein, fat, and carbohydrate. Although the intake of a normal diet decreased in normal rats that ingested HFCS, large amounts of fructose and glucose in HFCS promoted the generation of TAGE in liver cells. Serum TAGE levels and intracellular TAGE levels in the liver showed a positive correlation in rats, and, thus, serum TAGE levels have potential as a novel biomarker for the prevention of NAFLD in humans. 

## Figures and Tables

**Figure 1 nutrients-11-01612-f001:**
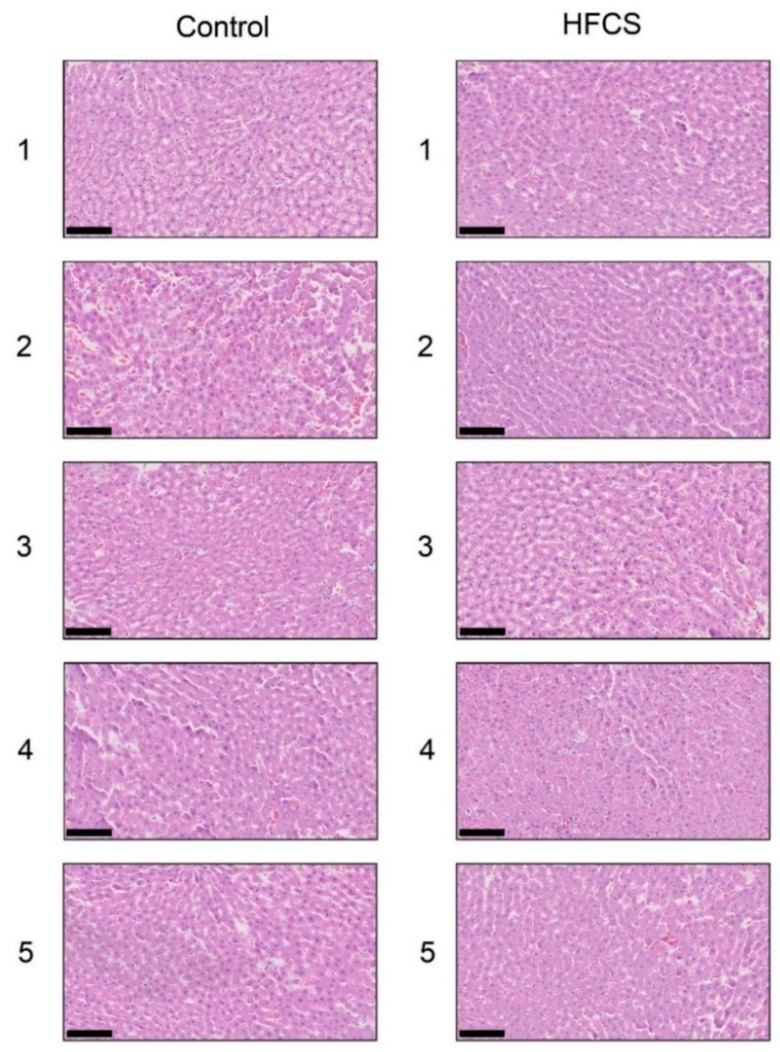
Histological analysis of the liver with hematoxylin and eosin staining. Control; control group, HFCS; HFCS group. The numbers 1–5 indicate each rat in the control group (*n* = 5) or HFCS group (*n* = 5). The scale bar represents 100 μm.

**Figure 2 nutrients-11-01612-f002:**
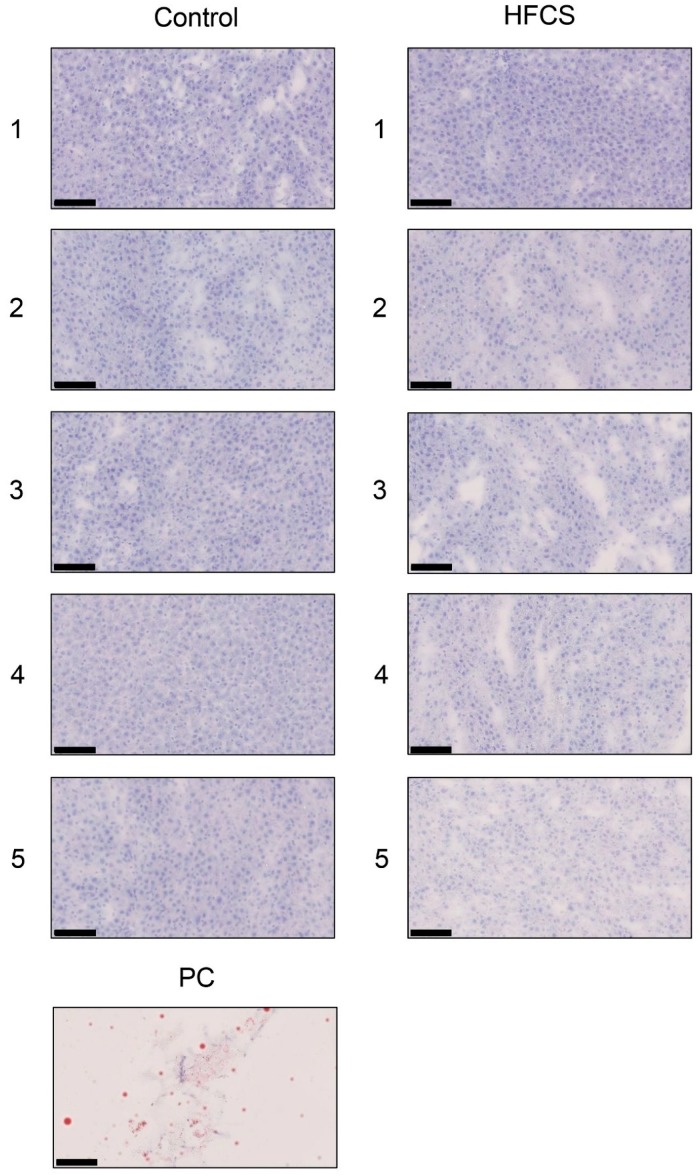
Histological analysis of the liver with Oil Red O staining. Control; control group, HFCS; HFCS group. The numbers 1–5 indicate each rat in the control group (*n* = 5) or HFCS group (*n* = 5). PC; Positive control. Adipose tissue of rat. Fat-positive areas stained red in cells. The scale bar represents 100 μm.

**Figure 3 nutrients-11-01612-f003:**
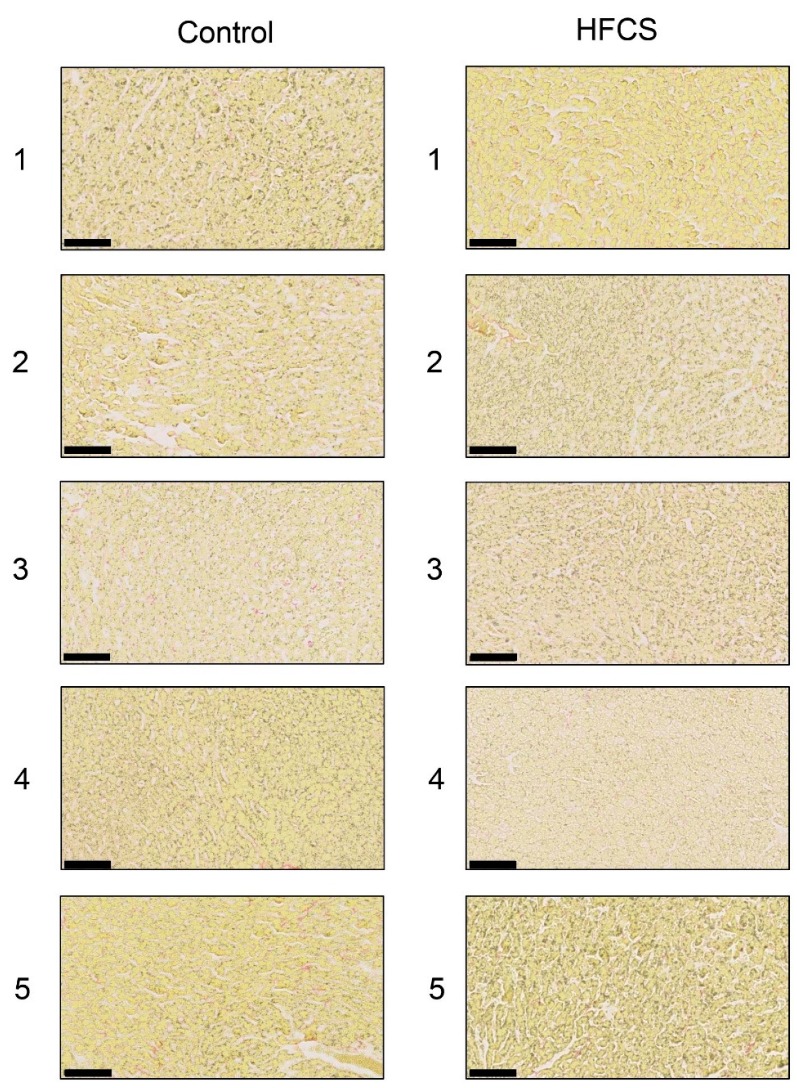
Histological analysis of the liver with Sirius Red staining. Control; control group, HFCS; HFCS group. The numbers 1–5 indicate each rat in the control group (*n* = 5) or HFCS group (*n* = 5). Type I or III collagen-positive areas stained red in cells. Muscle fiber and cytoplasm areas stained yellow. The scale bar represents 100 μm.

**Figure 4 nutrients-11-01612-f004:**
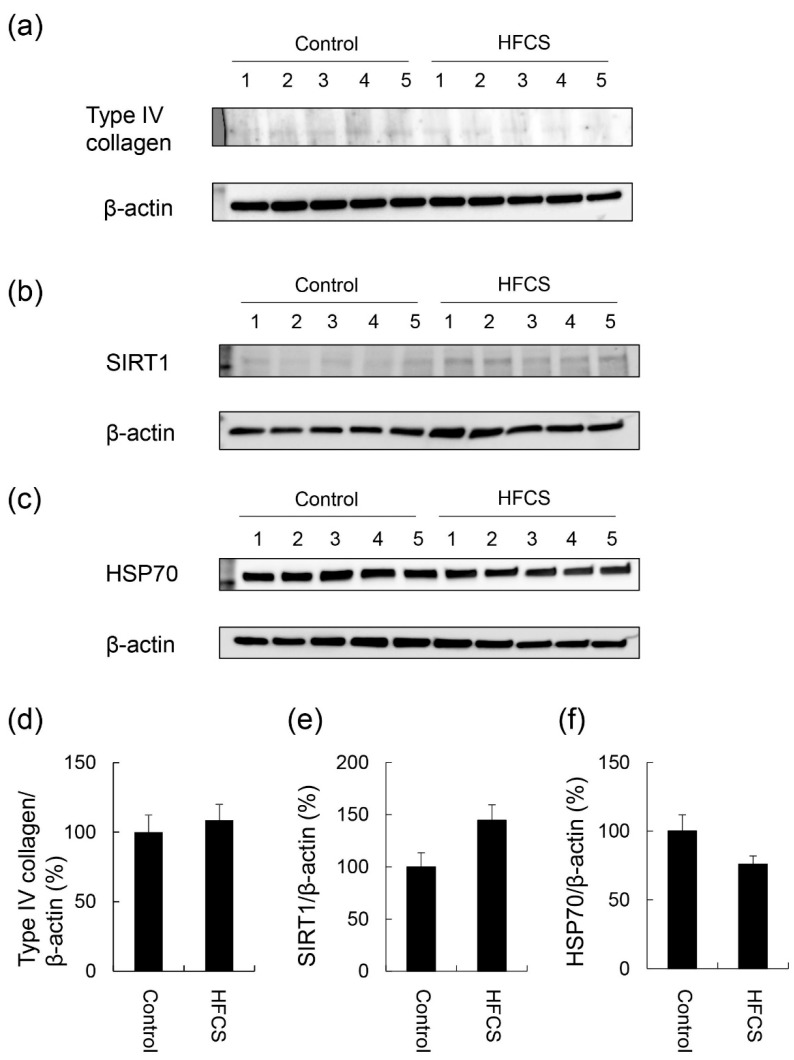
Expression of type IV collagen, SIRT1, and HSP70 in the liver. Control; control group (*n* = 5), HFCS; HFCS group (*n* = 5). The numbers 1–5 indicate each rat in the control or HFCS group. Tissue lysates (15 μg of protein/lane) were loaded on 4–15% gradient polyacrylamide gels. Proteins on polyvinylidene difluoride (PVDF) membranes were probed with anti-type IV collagen, anti-SIRT1, anti-HSP70, and anti-β-actin antibodies. (**a**) Type IV collagen and β-actin bands were analyzed with Western blotting (WB). (**b**) SIRT1 and β-actin bands were analyzed with WB. (**c**) HSP70 and β-actin bands were analyzed with WB. (**d**) The expression of type IV collagen in the control and HFCS groups was normalized with β-actin. (**e**) The expression of SIRT1 in the control and HFCS groups was normalized with β-actin. (**f**) The expression of HSP70 in the control and HFCS groups was normalized with β-actin. (**a**–**c**) β-actin was used as the loading control. (**d**–**f**) Data are shown as means ± S.E. (*n* = 5). *p*-Values were based on the Mann-Whitney U-test.

**Figure 5 nutrients-11-01612-f005:**
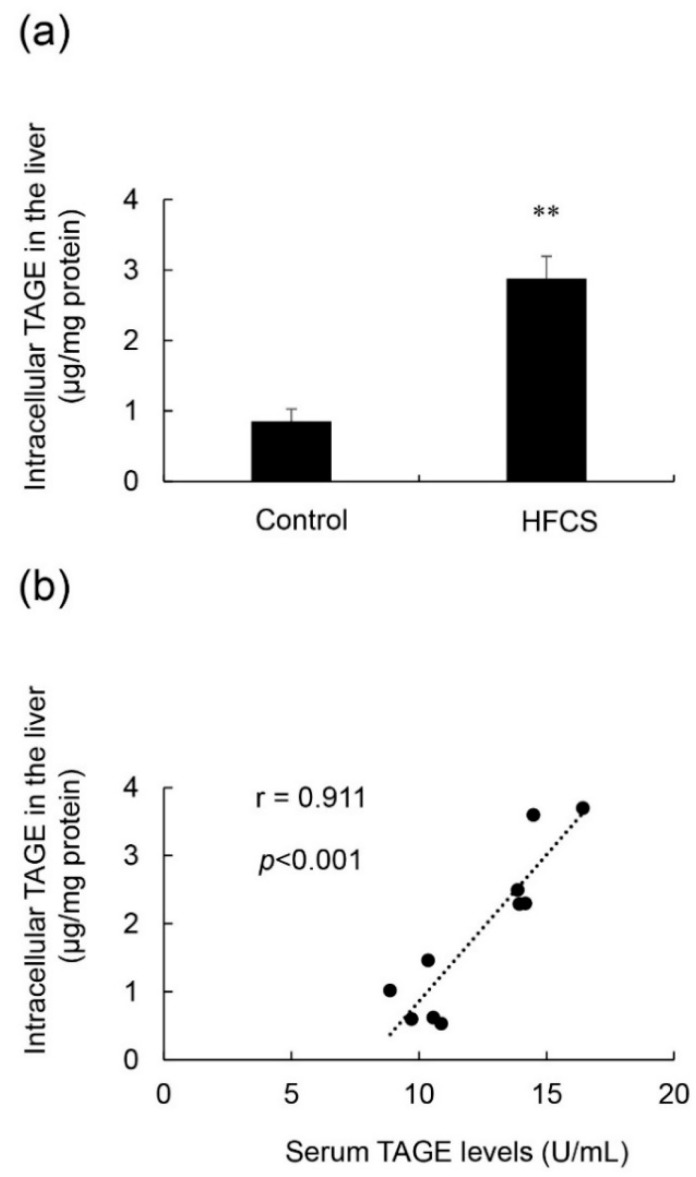
Calculation of intracellular TAGE levels in the liver, and the relationship between intracellular TAGE and serum TAGE levels. (**a**) Control; control group (*n* = 5), HFCS; HFCS group (*n* = 5). Tissue lysates (10 μg of protein/lane) were loaded on the PVDF membrane, and proteins were probed with the anti-TAGE antibody or neutralized anti-TAGE antibody. The amount of TAGE was calculated based on a standard curve for TAGE-BSA. Data are shown as means ± S.E. (*n* = 5). *p*-Values were based on the Mann-Whitney U-test. ** *p* < 0.01 vs. control. (**b**) Relationship between intracellular TAGE levels in the liver and serum levels of TAGE (*n* = 10).

**Figure 6 nutrients-11-01612-f006:**
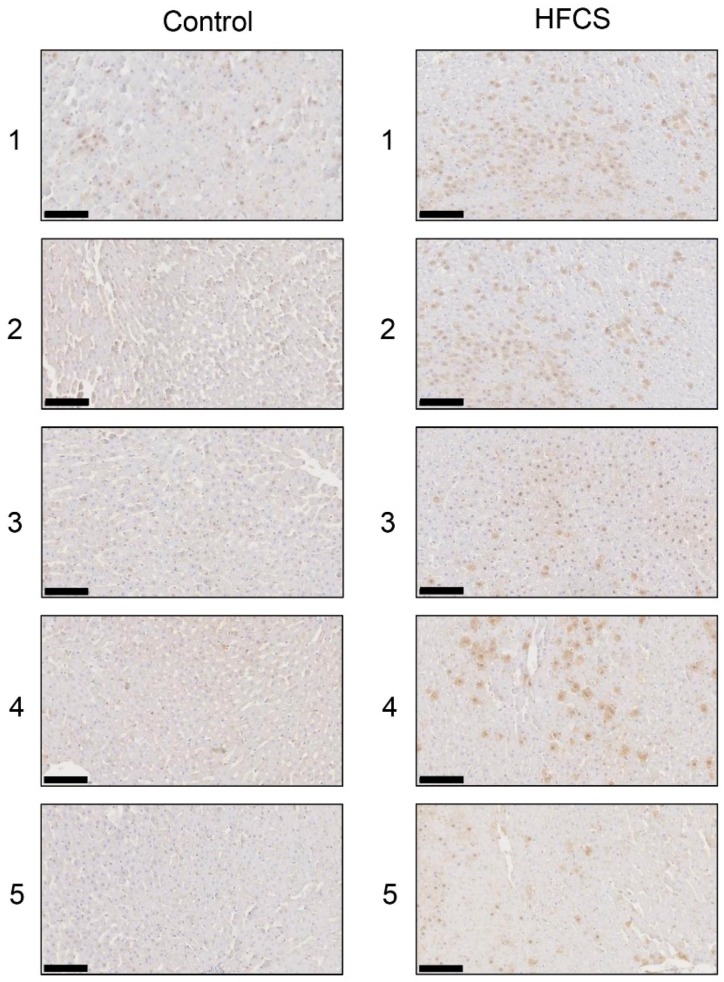
Immunostaining of intracellular TAGE in the liver. The numbers 1–5 indicate each rat in the control group (*n* = 5) or HFCS group (*n* = 5). Control; control group, HFCS; HFCS group. TAGE-positive areas stained brown in cells. The scale bar represents 100 μm.

**Figure 7 nutrients-11-01612-f007:**
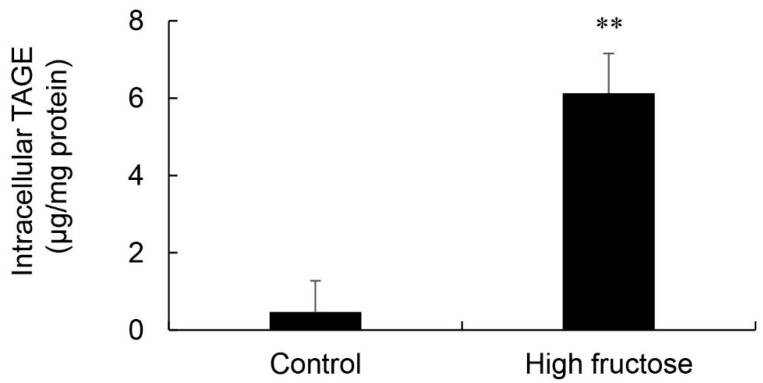
Calculation of intracellular TAGE levels in Sprague Dawley rat primary hepatocytes. Control; control medium (*n* = 4), High fructose; high fructose medium (*n* = 4). Cell lysates (4.0 μg of protein/lane) were loaded on PVDF membranes, and proteins were probed with the anti-TAGE antibody or neutralized anti-TAGE antibody. The amount of TAGE was calculated based on a standard curve for TAGE-BSA. Data are shown as means ± S.D. (*n* = 4). *p*-Values were based on the Mann-Whitney U-test. ** *p* < 0.01 vs. control.

**Figure 8 nutrients-11-01612-f008:**
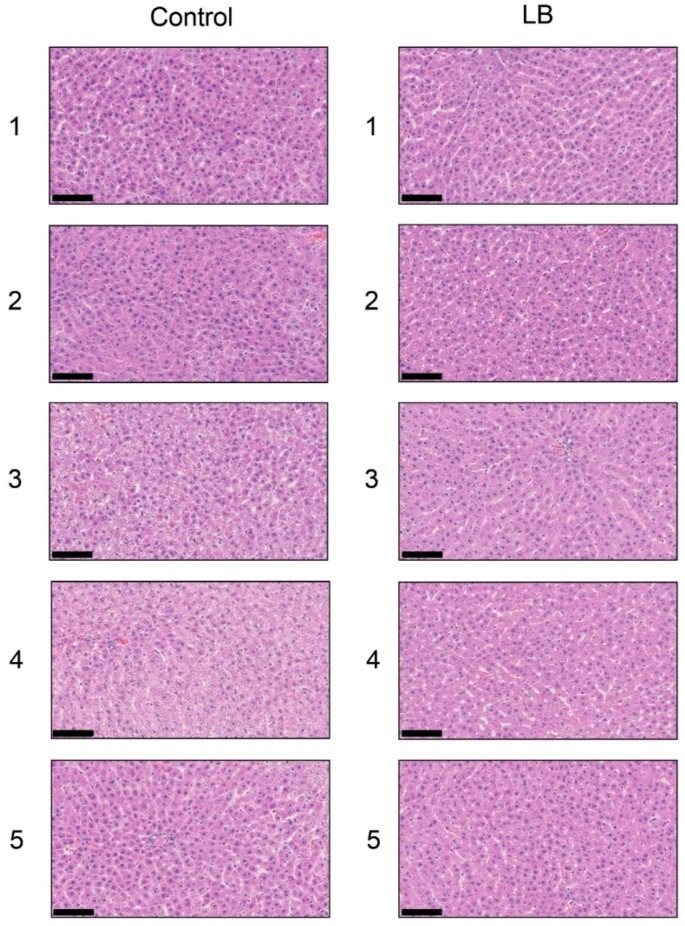
Histological analysis of the liver with hematoxylin and eosin staining. The numbers 1–5 indicate each rat in the control group (*n* = 5) or *Lactobacillus* group (*n* = 5). Control; control group, LB; *Lactobacillus* beverage group. The scale bar represents 100 μm.

**Figure 9 nutrients-11-01612-f009:**
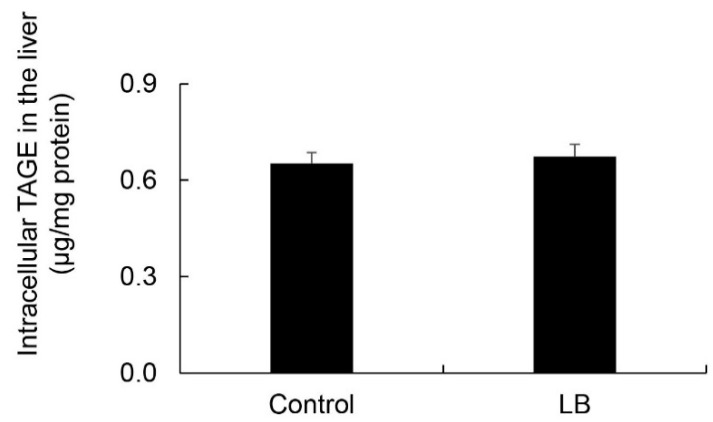
Calculation of intracellular TAGE levels in the liver. Control; control group (*n* = 5), LB; *Lactobacillus* beverage group (*n* = 5). Tissue lysates (10 μg of protein/lane) were loaded on PVDF membranes, and proteins were probed with the anti-TAGE antibody or neutralized anti-TAGE antibody. The amount of TAGE was calculated based on a standard curve for TAGE-BSA. Data are shown as means ± S.E. (*n* = 5). *p*-Values were based on the Mann-Whitney U-test.

**Table 1 nutrients-11-01612-t001:** Intake parameters of the normal diet and high fructose corn syrup (HFCS) beverage and body and liver weights.

Parameters	Control (*n* = 5)	HFCS (*n* = 5)
Normal diet (g)	16.9 ± 0.5	12.0 ± 0.4 **
Crude protein in the normal diet (g)	4.2 ± 0.1	3.0 ± 0.1 **
Crude fat in the normal diet (g)	0.85 ± 0.03	0.60 ± 0.02 **
NFE in the normal diet (g)	8.4 ± 0.2	6.0 ± 0.2 **
HFCS beverage (mL)		127.4 ± 5.6
Fructose from the HFCS beverage (g)		7.2 ± 0.3
Glucose from the HFCS beverage (g)		5.9 ± 0.3
Energy from the normal diet (kcal)	58.1 ± 1.7	41.3 ± 1.4 **
Energy from the HFCS beverage (kcal)		48.2 ± 2.1
Body weight (g)	496.0 ± 8.7	527.8 ± 14.4
Liver weight (g)	15.6 ± 0.5	18.0 ± 0.8
Liver index (g/kg)	31.5 ± 0.6	34.0 ± 0.7 *

Control; control group, HFCS (high-fructose corn syrup); HFCS group. NFE; nitrogen-free extracts. Liver index; liver weight/body weight (g/kg). Data are shown as means ± S.E. (*n* = 5). *p*-Values were based on the Mann-Whitney U-test. * *p* < 0.05, ** *p* < 0.01 vs. control.

**Table 2 nutrients-11-01612-t002:** Serum and plasma biochemistries in the HFCS group.

Parameters	Control (*n* = 5)	HFCS (*n* = 5)
TAGE (U/mL)	10.08 ± 0.36	14.58 ± 0.47 **
Glucose (mM)	9.70 ± 0.51	14.08 ± 0.73 **
Glycoalbumin (%)	1.30 ± 0.22	1.70 ± 0.18
BUN (mM)	8.92 ± 0.32	4.29 ± 0.54 **
UA (μM)	197.47 ± 10.02	222.46 ± 8.33
Ca (mM)	2.64 ± 0.04	2.82 ± 0.01
AST (IU/L)	134.4 ± 19.3	88.2 ± 7.2
ALT (IU/L)	72.0 ± 6.0	41.2 ± 2.2 **
LDH (IU/L)	944.8 ± 143.3	780.6 ± 89.9
CK (IU/L)	910.2 ± 174.0	653.2 ± 74.7
T-CHO (mM)	2.42 ± 0.08	2.73 ± 0.19
TG (mM)	0.79 ± 0.13	1.61 ± 0.36 *
LDL-C (mM)	0.22 ± 0.02	0.20 ± 0.01
T-BIL (μM)	0.68 ± 0.05	0.82 ± 0.10

Control; control group, HFCS; HFCS group. TAGE; toxic advanced glycation end-products, BUN; blood urea nitrogen, UA; uric acid, Ca; calcium, AST; aspartate aminotransferase, ALT; alanine aminotransferase, LDH; lactate dehydrogenase, CK; creatinine kinase, T-CHO; total cholesterol, TG; triglyceride, LDL-C; low-density lipoprotein cholesterol, T-BIL; total bilirubin. Data are shown as means ± S.E. (*n* = 5). *p*-Values were based on the Mann-Whitney U-test. * *p* < 0.05, ** *p* < 0.01 vs. control.

**Table 3 nutrients-11-01612-t003:** NAFLD activity scores in control and HFCS groups.

Group	*N*	Steatosis	Inflammation	Ballooning	Score
		0 1 2 3	0 1 2 3	0 1 2	
Control	5	5 0 0 0	5 0 0 0	5 0 0	0
HFCS	5	5 0 0 0	5 0 0 0	5 0 0	0

NAFLD activity scores (NAS): The unweighted sum of steatosis, lobular inflammation, and hepatocellular ballooning. Control; control group, HFCS; HFCS group.

**Table 4 nutrients-11-01612-t004:** Intake parameters of the normal diet and *Lactobacillus* beverage and body and liver weights.

Parameters	Control (*n* = 5)	LB (*n* = 5)
Normal diet (g)	19.9 ± 0.7	10.3 ± 0.6 **
Crude protein in the normal diet (g)	5.0 ± 0.2	2.6 ± 0.2 **
Crude fat in the normal diet (g)	0.87 ± 0.03	0.45 ± 0.03 **
NEF in the normal diet (g)	9.9 ± 0.4	5.1 ± 0.3 **
*Lactobacillus* beverage (mL)		54.8 ± 3.8
Crude protein in the *Lactobacillus* beverage (g)		0.4 ± 0.0
Crude fat in the *Lactobacillus* beverage (g)		0.08 ± 0.01
Fructose in the *Lactobacillus* beverage (g)		2.8 ± 0.2
Glucose in the *Lactobacillus* beverage (g)		3.0 ± 0.2
Sucrose in the *Lactobacillus* beverage (g)		2.3 ± 0.2
Lactose in the *Lactobacillus* beverage (g)		0.8 ± 0.1
Other carbohydrates in the *Lactobacillus* beverage (g)		0.8 ± 0.1
Energy from the normal diet (kcal)	67.4 ± 2.5	35.0 ± 2.1 **
Energy from the *Lactobacillus* beverage (kcal)		42.2 ± 2.6
Body weight (g)	478.8 ± 14.9	481.8 ± 8.9
Liver weight (g)	14.3 ± 0.4	14.8 ± 0.5
Liver index (g/kg)	29.9 ± 0.4	30.8 ± 0.8

Control; control group, LB; *Lactobacillus* beverage group. NFE; nitrogen-free extracts. Liver index; liver weight/body weight (g/kg). Data are shown as means ± S.E. (*n* = 5). *p*-Values were based on the Mann-Whitney U-test. ** *p* < 0.01 vs. control.

**Table 5 nutrients-11-01612-t005:** Serum and plasma biochemistries in the *Lactobacillus* group.

Parameters	Control (*n* = 5)	LB (*n* = 5)
TAGE (U/mL)	10.12 ± 0.22	10.02 ± 0.83
Glucose (mM)	9.73 ± 0.41	14.65 ± 1.42 **
Glycoalbumin (%)	1.78 ± 0.16	1.34 ± 0.43
BUN (mM)	9.77 ± 0.44	6.34 ± 0.63 **
UA (μM)	173.68 ± 12.67	205.80 ± 25.56
Ca (mM)	2.64 ± 0.02	2.70 ± 0.01
AST (IU/L)	142.2 ± 6.5	109.6 ± 25.9
ALT (IU/L)	70.6 ± 8.0	69.4 ± 29.7
LDH (IU/L)	831.6 ± 69.9	1071.2 ± 310.4
CK (IU/L)	917.8 ± 69.3	900.2 ± 122.5
T-CHO (mM)	1.96 ± 0.15	2.45 ± 0.27
TG (mM)	0.73 ± 0.17	1.04 ± 0.25
LDL-C (mM)	0.17 ± 0.01	0.15 ± 0.01
T-BIL (μM)	0.51 ± 0.05	0.55 ± 0.10

Control; control group, LB; *Lactobacillus* beverage group. TAGE; toxic advanced glycation end-products, BUN; blood urea nitrogen, UA; uric acid, Ca; calcium, AST; aspartate aminotransferase, ALT; alanine aminotransferase, LDH; lactate dehydrogenase, CK; creatinine kinase, T-CHO; total cholesterol, TG; triglyceride, LDL-C; low-density lipoprotein cholesterol, T-BIL; total bilirubin. Data are shown as means ± S.E. (*n* = 5). *p*-Values were based on the Mann-Whitney U-test. ** *p* < 0.01 vs. control.

**Table 6 nutrients-11-01612-t006:** NAFLD activity scores in control and *Lactobacillus* beverage groups.

Group	*N*	Steatosis	Inflammation	Ballooning	Score
		0 1 2 3	0 1 2 3	0 1 2	
Control	5	5 0 0 0	5 0 0 0	5 0 0	0
LB beverages	5	5 0 0 0	5 0 0 0	5 0 0	0

NAFLD activity scores (NAS): The unweighted sum of steatosis, lobular inflammation, and hepatocellular ballooning. Control; control group, LB beverage; *Lactobacillus* beverage group.

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
