# Peer review of "Evidence for Toxic Advanced Glycation End-Products Generated in the Normal Rat Liver"

_nutrients, 2019, doi:10.3390/nu11071612_

Round 1

Reviewer 1 Report

The presented manuscript presents an under explored area of TAGE in liver. The presence of TAGE has gathered much needed attention during recent years. Although the current manuscript offers an interesting data however the manuscript writing quality can be improved hugely. Overall

1.      For the histology data, adding histological scoring will improve the interpretation of the data. Authors need to discuss potential reason why there are no histological differences in the treated vs control. The current data outcome is different than the previous literature data and does warrant a discussion at least.

2.      In the result section authors started with SIRT1 and HSP70 levels. The writing part in this section can be improved. Authors did discuss in discussion section but the writing in result section seemed abrupt as a reader until we read discussion. Similar thing for lactobacillus section.

3.      Addition of some population data supporting authors claim will help create  a better impact of the study.

Author Response

We would like to thank both of the reviewers for their comments on our manuscript. We addressed the comments provided and hope that our responses are satisfactory. Our point-by-point responses to each reviewer are listed below.

Point-by-point responses to the comments of Reviewer #1

Reviewer comment #1

1.        For the histology data, adding histological scoring will improve the interpretation of the data. Authors need to discuss potential reason why there are no histological differences in the treated vs control. The current data outcome is different than the previous literature data and does warrant a discussion at least.

Response: We provided data on NAFLD activity scores (NAS). We added sentences on NAS to the Results section on page 6 (Line 257-258) and page 7 (Line 294-296). We provided new Table 3 (page 16) and new Table 6 (page 18) showing NAS data.

We also discussed potential reasons for the lack of histological differences between the treated vs control groups. In the original manuscript, we suggested that intracellular TAGE generated in the liver in the HFCS group were not sufficient to induce inflammation and fibrosis. We described this on page 20 (Line 555-557). Furthermore, we discussed why fat accumulation was not detected and steatosis was not induced. The accumulation of fat in the liver is affected by dietary saccharides and fats (new Reference 36). In the present study, the HFCS group gained 0.7-fold the amount of fat of the control group. Similarly, the Lactobacillus beverage group gained 0.6-fold the amount of fat of the control group. This lack of dietary fat may be one of the reasons why steatosis was not induced. We added sentences to the Discussion section on page 20, Line 565-page 21, Line 578 and page 21 (Line 610-612).

Since inflammation and fibrosis were not detected in the liver in the Lactobacillus beverage group, intracellular TAGE generation did not appear to be sufficient to induce inflammation and fibrosis. We added sentences to the Discussion section on page 21 (Line 597-604).

2.        In the result section authors started with SIRT1 and HSP70 levels. The writing part in this section can be improved. Authors did discuss in discussion section but the writing in result section seemed abrupt as a reader until we read discussion. Similar thing for lactobacillus section.

Response: We added the title ‘Expression patterns of proteins associated with steatosis, inflammation, and fibrosis’ for the Western blot analysis of type IV collagen, SIRT1, and HSP70 to the Results section on page 6 (Line 262).

We prepared an HFCS beverage that simulated HFCS-containing beverages that are commercially available. On the other hand, the Lactobacillus beverage is a commercial drink. To explain the difference between the HFCS and Lactobacillus beverages, we added sentences to the Introduction section on page 2 (Line 51-53).

3.        Addition of some population data supporting authors claim will help create a better impact of the study.

Response: We added data on Oil Red O staining and Sirius Red staining for fat and fibrosis in the liver in the HFCS and Lactobacillus beverage groups, as well as that on H.E. staining and immunostaining of the anti-TAGE antibody in the liver in the Lactobacillus beverage group. We added sentences to the Results section on page 6 (Line 259-261), page 7 (Line 291-298, Line 305-308), and the Discussion section on page 19 (Line 509-521), page 20 (Line 532-533), and page 21 (Line 597-598). We provided new Figures 2, 3, and 8 and new Figures S4, S5, and S6 on page 9, 10, 14, 26, 27, and 28. We also described previous reference 19 and inserted new reference 34 to explain the method for Oil Red O staining. We inserted new references 34 and 35 to explain the method for Sirius Red staining.

We analyzed the expression of type IV collagen with Western blotting (WB). We described why we selected type IV collagen in the Discussion section on page 19, Line 514-page20, Line 522 and inserted new references 37 and 38.

Although we need to analyze the expression of type I and III collagen with WB, we were unable to achieve this by the deadline of the revision. However, we explained that we analyzed type I and III collagens with Sirius Red staining. We added a sentence to the Discussion section on page 20 (L519-521).

Reviewer 2 Report

Takata et al. in this manuscript found that large amounts of fructose and glucose promoted the generation of toxic advanced glycation end-products (TAGE) in the liver cells. In particular, serum TAGE levels and intracellular hepatic TAGE levels both increased in the high-fructose corn syrup (HFCS) group. A positive correlation was observed between intracellular TAGE levels in the liver and serum TAGE levels. On the other hand, rats that drank Lactobacillus beverage, a commercial drink that contains glucose, fructose, and sucrose did not show any increases in intracellular TAGE or serum TAGE levels. The production of TAGE was promoted by HFCS, which may increase the risk of NAFLD. The paper examines an interesting topic but it has strong limitations. Authors need to address some important concerns listed below and make stronger their findings.

Major comments:

1.       Authors conclude that serum TAGE levels and intracellular TAGE levels in the liver showed a positive correlation in rats, and, thus, serum TAGE levels have potential as a novel biomarker for the prevention of NAFLD in humans. However, they did not observe any NAFLD phenotype in histological staining (H&E). Then, there is no correlation between TAGE levels and NAFLD phenotype. Authors should run further stainings on the liver tissues as Sirius Red for fibrosis or Oil Red O for fat content.

2.       In order to understand whether markers involved in fibrosis are upregulated also western blotting analysis of collagen could be tested.

3.       Authors performed staining (H&E), immunostaining (anti-TAGE) and intracellular quantification of TAGE for samples from HFCS group. On the hand, they show only TAGE quantification for samples from Lactobacillus group. Do authors have other data on Lactobacillus group?

4.       Authors should clarify and discuss better the difference in the results obtained with HFCS or Lactobacillus diet.

5.       How do the authors explain the fact that they do not see any fat accumulation in the liver sections after HFCS diet?

Minor comments:

1.       Authors should check the sentence on page 7 line 270-272.

Author Response

 We would like to thank both of the reviewers for their comments on our manuscript.

We addressed the comments provided and hope that our responses are satisfactory.

Our point-by-point responses to each reviewer are listed in the "Word file".

We would like to thank both of the reviewers for their comments on our manuscript. We addressed the comments provided and hope that our responses are satisfactory. Our point-by-point responses to each reviewer are listed below.

Point-by-point responses to the comments of Reviewer #2

Reviewer comment #2

1.       Authors conclude that serum TAGE levels and intracellular TAGE levels in the liver showed a positive correlation in rats, and, thus, serum TAGE levels have potential as a novel biomarker for the prevention of NAFLD in humans. However, they did not observe any NAFLD phenotype in histological staining (H&E). Then, there is no correlation between TAGE levels and NAFLD phenotype. Authors should run further stainings on the liver tissues as Sirius Red for fibrosis or Oil Red O for fat content.

Response: We stained liver tissues with Sirius Red for fibrosis and Oil Red O for fat content. These stains were performed on the liver in the HFCS and Lactobacillus beverage groups. We added these sentences to the Results section on page 6 (Line 259-261) and page 7 (Line 296-298), and the Discussion section on page 19 (Line 509-513, Line 519-521), page 20 (Line 532-533), and page 21 (Line597-598). We provided new Figures 2, 3, Figures S4, and S5 on page 9, 10, 26 and 27. We also described previous reference 19 and inserted new reference 34 to explain the method for Oil Red O staining. We inserted new references 34 and 35 to explain the method for Sirius Red staining.

2.    In order to understand whether markers involved in fibrosis are upregulated also western blotting analysis of collagen could be tested.

Response: We analyzed the expression of type IV collagen with Western blotting (WB) against tissue samples collected from the HFCS group. We described why we selected type IV collagen in the Discussion section on page 19, Line 514-page 20, line 522 and inserted new references 37 and 38.

Although we need to analyze the expression of type I and III collagen with WB, we were unable to achieve this by the deadline of the revision. However, we explained that we analyzed type I and III collagens with Sirius Red staining.

We added a sentence to the Discussion section on page 19 (L519-521).

3.    Authors performed staining (H&E), immunostaining (anti-TAGE) and intracellular quantification of TAGE for samples from HFCS group. On the hand, they show only TAGE quantification for samples from Lactobacillus group. Do authors have other data on Lactobacillus group?

Response: We provided data on H.E. staining, Oil Red O staining, Sirius Red staining, and immunostaining of the anti-TAGE antibody in the Lactobacillus beverage group. We added sentences to the Results section on page 7 (Line 291-298, Line 305-308), and the Discussion section on page 21 (Line 597-598). We also provided new Figure 8, new Figures S4-6, and new Table 6 on page 14, 18, 26, 27, and 28.

4.       Authors should clarify and discuss better the difference in the results obtained with HFCS or Lactobacillus diet.

Response: As reviewer #2 pointed out, we need to clarify and discuss the differences in the results obtained with the HFCS and Lactobacillus diets.

The HFCS beverage increased the liver index, increased triglycerides (TG), and decreased alanine aminotransferase (ALT) activity, whereas the Lactobacillus beverage did not. We considered one of the reasons to be a component of the Lactobacillus beverage. We added sentences to the Discussion section on page 21 (Line 590-596).

The HFCS beverage induced an increase in intracellular TAGE in the liver, whereas the Lactobacillus beverage did not. We suggested that a greater intake of fructose/glucose results in the generation of intracellular TAGE. We indicated that serum levels of TAGE may not have increased because the generation of intracellular TAGE was not promoted. Moreover, we discussed the possibility that other components of the Lactobacillus beverage or Lactobacillus inhibited the generation of intracellular TAGE or promoted the degradation of intracellular TAGE.

We added sentences to the Discussion section on page 21 (Line 599-608).

5.    How do the authors explain the fact that they do not see any fat accumulation in the liver sections after HFCS diet?

Response: We suggested a reason for why fat accumulation was not observed in liver sections in the HFCS group. The accumulation of fat in the liver is affected by dietary saccharides and fats (new reference 36). In the present study, the HFCS group gained 0.7-fold the amount of fat of the control group. We added sentences to the Discussion section on page 20 Line 565-page 21, Line 578.

Similarly, the Lactobacillus beverage group gained 0.6-fold the amount of fat of the control group. This lack of dietary fat may be one of the reasons why steatosis was not induced. We added sentences to the Discussion section on page 21 (Line 610-612).

Minor comments:

1.    Authors should check the sentence on page 7 line 270-272.

Response: We revised the sentence on page 7 Line 273-278 (on page 7, Line 270-272 in the previous manuscript).

Round 2

Reviewer 1 Report

The authors have answered all my doubts. 

Reviewer 2 Report

The authors have addressed all of my comments.